# Gait Environment Recognition Using Biomechanical and Physiological Signals with Feed-Forward Neural Network: A Pilot Study

**DOI:** 10.3390/s25144302

**Published:** 2025-07-10

**Authors:** Kyeong-Jun Seo, Jinwon Lee, Ji-Eun Cho, Hogene Kim, Jung Hwan Kim

**Affiliations:** 1Department of Rehabilitation & Assistive Technology, National Rehabilitation Center, Ministry of Health and Welfare, Seoul 01022, Republic of Korea; 2Department of Industrial and Management Engineering, Gangneung-Wonju National University, Wonju 26403, Gangwon-do, Republic of Korea; 3Department of Mechanical Engineering, University of Michigan, Ann Arbor, MI 48109, USA; 4Department of Rehabilitation Medicine, Ewha Womans University & Medical Center Mokdong Hospital, Seoul 07985, Republic of Korea

**Keywords:** feed-forward neural network (FFNN), gait environment, wearable sensor, multimodal sensor

## Abstract

Gait, the fundamental form of human locomotion, occurs across diverse environments. The technology for recognizing environmental changes during walking is crucial for preventing falls and controlling wearable robots. This study collected gait data on level ground (LG), ramps, and stairs using a feed-forward neural network (FFNN) to classify the corresponding gait environments. Gait experiments were performed on five non-disabled participants using an inertial measurement unit, a galvanic skin response sensor, and a smart insole. The collected data were preprocessed through time synchronization and filtering, then labeled according to the gait environment, yielding 47,033 data samples. Gait data were used to train an FFNN model with a single hidden layer, achieving a high accuracy of 98%, with the highest accuracy observed on LG. This study confirms the effectiveness of classifying gait environments based on signals acquired from various wearable sensors during walking. In the future, these research findings may serve as basic data for exoskeleton robot control and gait analysis.

## 1. Introduction

Gait is a fundamental human movement essential for daily activities, involving complex interactions among the nervous and musculoskeletal systems [1,2,3]. In real-life environments, individuals navigate level ground, ramps, and stairs across various surfaces. Understanding gait in these diverse conditions is crucial for evaluating mobility and functional capacity.

Gait analysis provides quantitative insights into human locomotion and is widely used in health assessment, rehabilitation, and the diagnosis of gait disorders [4,5]. Variations in gait patterns under different surface conditions may serve as early indicators of neurological disorders such as stroke and Parkinson’s disease [6,7,8,9,10]. For older adults and individuals with disabilities, navigating slopes or stairs presents elevated fall risks, which can be mitigated through early detection and intervention [11,12,13].

Different walking surfaces impose distinct biomechanical demands and motor strategies. While level-ground walking is relatively stable, ramps and stairs require greater postural control and joint torque, particularly at the ankle, hip, and knee [14,15,16,17]. These differences underscore the need to classify walking environments for targeted functional evaluation and personalized rehabilitation.

Prior studies have employed diverse methodological approaches to reflect varying walking environments, considering the unique characteristics and risk levels of the study populations. Most studies included basic walking terrains commonly encountered in daily life, such as LG, ramps, and stairs, to ensure ecological validity [18,19,20,21,22,23]. Beyond standard terrains, various studies have incorporated additional surface types such as sand, gravel, grass, forest trails, and uneven surfaces, better simulating the complexity of community walking environments [24,25,26,27,28,29]. This accurately reflects gait and fall risks in real-world environments, highlighting the need for quantitative classification and recognition technologies for walking conditions, as indicated in this study.

Historically, walking environment classification has primarily relied on inertial measurement unit (IMU)-based approaches. To enhance measurement accuracy, most studies have employed multiple IMUs attached to the thighs, shanks, and feet to acquire kinematic data, including acceleration and angular velocity during gait [8,18,23,24,25]. Sensor placement has been shown to significantly affect the performance of IMU-based classification, with common locations including the lower limbs such as the shank, thigh, and foot [30]. Some studies have proposed models using a single IMU to classify or estimate the type of walking environment [21,29,31]. Practical approaches for recognizing walking environments have recently been explored using built-in smartphone IMUs [27].

Some studies attached multiple surface electromyography (sEMG) sensors to the legs [19] or combined them with IMUs to classify walking environments effectively by integrating muscle activity with biomechanical data [20]. sEMG measures the electrical activity of muscles using skin-surface electrodes, enabling the detection of physiological responses associated with muscle contractions during gait. These signals reflect muscle activation levels and are valuable for assessing the physiological behavior of leg muscles during locomotion, particularly by capturing the timing, intensity, and coordination of muscular contractions. EMG signals provide detailed insights into neuromuscular control and are frequently used in gait analysis, rehabilitation engineering, and prosthetic device development [32,33,34,35].

Smart insoles, another class of wearable devices, are designed to measure the ground reaction force (GRF) on the plantar surface of the foot during gait. When integrated with IMUs, sEMG, and position sensors, smart insoles enhance the accuracy of classifying walking environments [20].

A broad spectrum of algorithms, from traditional machine learning (ML) to deep learning (DL) techniques, has been applied to classify walking environments. Classical classification algorithms in ML, such as k-nearest neighbors, decision trees, support vector machines, and Random Forests, have been widely used to distinguish between different gait patterns and human activities [24,25,27]. DL studies have employed architectures, including convolutional neural networks (CNNs), long short-term memory (LSTM), Global-LSTM, and artificial neural networks (ANNs) [18,19,22,24,26,28,29,31].

Despite these advancements, existing studies have relied on specific sensors or their combinations. While this approach effectively captures kinematic information related to gait, it often overlooks psychological and physiological responses to environmental or individual changes. Furthermore, DL models with high complexity and dimensionality, such as CNN and LSTM networks, are frequently employed to enhance the classification accuracy. However, these models are often unsuitable for wearable systems requiring real-time processing or operating in low-resource environments.

This study integrates IMUs, smart insoles, and galvanic skin response (GSR) sensors to collect multidimensional gait data that encompasses kinematic, biomechanical, and physiological signals, thereby addressing these limitations. Using the collected data, this study aims to classify walking environments using a simple feed-forward neural network (FFNN) with a single hidden layer—a lightweight architecture that ensures computational efficiency while balancing accuracy and practical applicability.

The scholarly contributions of this study are as follows: First, while previous studies relied solely on IMUs or limited sensor combinations that included only a few biomechanical sensors, this study simultaneously leveraged IMUs, smart insoles, and GSR sensors to collect multidimensional data, including kinematic, biomechanical, and physiological signals, for gait environment classification. No study has attempted to integrate photoplethysmogram (PPG) signals from GSR sensors for classifying walking environments. This novel application provides a basis for assessing the applicability of physiological responses in this context. Furthermore, this research provides a foundation for multilevel interpretations of bodily responses to variations in walking environments, which may support the development of practical classification systems for future applications. Second, a simple neural network achieves high classification performance, offering a more computationally efficient alternative to modern ML algorithms that typically require greater complexity and processing power. This demonstrates the effectiveness of lightweight models, providing a technological foundation for applications in practical systems such as wearable robots. Third, integrating multimodal sensor data enables feedback and user-adaptive control mechanisms—capabilities essential for designing control systems in exoskeletons and mobility-assistive devices. These systems can thereby more accurately recognize and regulate the walking environment of the user.

The remainder of this paper is organized as follows. Section 2 outlines the data collection process. Section 3 details the preprocessing methods and implementation of the ML algorithms. Section 4 presents the performance evaluation results and discusses key findings. Section 5 addresses study limitations and suggests future research directions.

## 2. Data Collection and Experimental Design

### 2.1. Participants

Data were collected from a total of five participants (four females and one male; mean age: 37.6 ± 6.5 years; height: 169.6 ± 8.6 cm; weight: 59.6 ± 11.5 kg). All participants were healthy adults with no history of neurological, musculoskeletal, or cardiovascular disorders. Each participant could walk independently across various indoor walking environments without using assistive devices. All participants were fully informed of the purpose and procedures of the study before participating and voluntarily provided their consent.

### 2.2. Sensor Setup

Figure 1 illustrates the sensor attachment locations used in this study and the types of data collected from each sensor. A Shimmer3 IMU (Shimmer Sensing, Dublin, Ireland) measured tri-axial acceleration and angular velocity. IMUs were attached to the shanks and insteps of both legs and the hypogastric region to collect data for analyzing trunk kinematics during gait. GRF data were collected using a smart insole (Loadsol, Novel GmbH, München, Germany). The vertical components of GRF applied to both feet were also measured. The smart insole was inserted into the regular footwear of the participant during movement. Finally, the Shimmer3 GSR sensor (Shimmer Sensing, Ireland) was attached to two fingers on one hand of each participant to measure PPG signals. PPG reflects autonomic nervous system activity, providing insights into physiological responses to various walking environments. All sensors were securely attached to avoid interference with the movement of the participants. Data from all devices were recorded at a sampling rate of 100 Hz.

### 2.3. Gait Environment

This study was conducted indoors within a typical building rather than a specialized laboratory. Figure 2 illustrates the gait environments used in the experiment: LG, descending ramp (DR), ascending ramp (AR), descending stairs (DS), and ascending stairs (AS). The LG was a flat, corridor-like surface with no elevation changes. The ramps were constructed at an inclination of approximately 5°, making them accessible to wheelchairs. The stairs complied with standard dimensional specifications. These environments were chosen to reflect realistic walking conditions while minimizing external factors like weather.

## 3. ML Framework

### 3.1. Data Collection and Preprocessing

All participants were instructed to walk at a natural, comfortable pace consistent with their usual walking speed. For the LG condition, they walked along a straight 10 m path. During the ramp condition, participants descended a 6 m ramp, paused briefly, then ascended it. The stair condition followed the same procedure; participants walked from the bottom to the top of the stairs, paused briefly, and then returned to the starting point. These procedures allowed for the collection of kinematic, biomechanical, and physiological data during gait.

The collected data underwent preprocessing for ML. First, sensor time axes were synchronized to align data temporally across all sources. The initial and final 1 s segments of each walking trial were excluded to remove unstable movements, leaving only the segment between 1 s after the start and 1 s before the end for analysis.

Data quality enhancement was performed using MATLAB R2024a (MathWorks, Inc., Natick, MA, USA) to reduce noise from sensor vibrations, body tremors, and transient signal spikes. First, a low-pass filter was applied to the triaxial acceleration and angular velocity data from the IMUs and the PPG signals from the GSR sensors, eliminating high-frequency noise [36,37]. Additionally, a Butterworth filter was used to denoise the GRF data [38]. Finally, the data were labeled according to their corresponding gait environment. Figure 3 illustrates the data processing procedure, which enables consistent extraction of data corresponding to the actual walking segments.

Table 1 presents the final dataset used for model training and evaluation. Due to partial data loss, DR samples were not available for Subject 3. The total number of samples per environment was imbalanced, with level-ground data being the most prevalent. Despite these limitations, the dataset retained sufficient variability across walking conditions and subjects to enable meaningful model training and evaluation.

### 3.2. FFNN

This study employed a multilayer FFNN to analyze the walking data. An FFNN comprises an input layer, one or more hidden layers, and an output layer, with information flowing unidirectionally from the input to the output without feedback loops.

The input data consisted of the extracted and preprocessed features derived from the IMU, smart insole, and GSR signals. Specifically, the input layer comprised three-dimensional acceleration and angular velocity data from the IMU, GRF data from the smart insole, and PPG signals from the GSR.

The hidden layers adopted a multilayer perceptron architecture with Rectified Linear Unit activation functions, enabling effective learning of the nonlinear characteristics of gait data. The numbers of hidden layers and nodes per layer were optimized via cross-validation. Additionally, dropout regularization was applied between hidden layers to prevent overfitting (Figure 4).

The output layer employed a softmax activation function to perform multiclass classification of the gait environments, including LG, ramps, and stairs. For multiclass classification, model training used the cross-entropy loss function defined as:(1)$$Loss_{cross-entropy}=-\sum_{i=1}^{n}t_{i}\log\left(p_{i}\right)$$
where n is the number of classes, $t_{i}$ is the truth label, and $p_{i}$ is the softmax probability of the ith class.

To evaluate the performance of the FFNN model, accuracy, precision, recall, and F1-score were calculated. Accuracy represents the proportion of correctly classified instances among all predictions made by the model, as defined in Equation (2).(2)$$Accuracy=\frac{TP+TN}{TP+FP+FN+TN}$$Precision refers to the proportion of correctly identified instances among all instances predicted as belonging to a given class. This indicates the reliability of the predictions of the model and is defined as(3)$$Precision=\frac{TP}{TP+FP}$$Recall represents the proportion of actual instances of a given class that the model correctly identifies, defined in Equation (4) as(4)$$Recall=\frac{TP}{TP+FN}$$The F1-score is the harmonic mean of precision and recall and is used as an evaluation metric that accounts for class imbalance.(5)$$F1-score=\frac{2\times Precision\times Recall}{Precision+Recall}$$
where true positive (TP) refers to the number of instances correctly classified as belonging to a specific class, false positive (FP) refers to the number of instances incorrectly classified as belonging to that class, false negative (FN) refers to the number of instances belonging to the class but misclassified as another class, and true negative (TN) refers to the number of instances correctly classified as not belonging to the target class.

This study collected 47,033 walking data samples for training under three different conditions (Cases 1 to 3), as illustrated in Figure 5. Of the dataset, 32,923 samples (70%) were used to train the FFNN model, while the remaining data were used to evaluate the generalization performance of the model via validation and testing. The dataset was shuffled and split without separating participant data.

## 4. Results and Discussion

### 4.1. Gait Environment Analysis

Table 2 summarizes the training results. In each case, classification accuracy improved as the number of hidden layer units increased, reaching its peak performance with 1000 units. The highest accuracies were 95.8%, 98.0%, and 97.5% for Cases 1, 2, and 3, respectively.

Table 3 presents confusion matrices for the three models alongside their accuracy, precision, recall, and F1-score. Among them, Case 2 demonstrated the best performance based on the macro F1-score. Performance across the gait environments was observed in the following order: LG, DS, AS, AR, and DR.

Among the three tested cases, Case 2 yielded the best classification performance across all gait environments. The model achieved a macro F1-score of 97.73%, with particularly high precision and recall in LG and stair-walking conditions (AS, DS). In contrast, Case 1 and Case 3 showed slightly lower overall F1-scores (95.83% and 97.50%, respectively), although all cases exceeded 95% in the macro F1-score, demonstrating the feasibility of gait environment classification using multimodal wearable sensors and a lightweight FFNN model.

### 4.2. Performance Comparison by Sensor Configuration

Based on the results presented earlier, this section aims to provide a detailed comparison and analysis of classification performance across different sensor combinations. The analysis was conducted using Case 2, which demonstrated the best performance, as the reference configuration. The sensor combinations were configured into the following three setups for the experiment: (1) IMU only, (2) IMU combined with a smart insole, and (3) IMU combined with a GSR sensor.

Table 4 presents the classification performance metrics for each sensor combination, including accuracy, precision, recall, and F1-score. Among the three tested configurations, the IMU + GSR setup yielded the highest performance, with an accuracy of 94.4% and an F1-score of 94.34%. The IMU + smart insole combination achieved a slightly lower performance, with an accuracy of 93.2% and an F1-score of 93.25%, while the IMU-only configuration showed the lowest performance across all metrics. The full multimodal setup (Case 2), which includes IMU, smart insole, and GSR, achieved the best overall results, with an accuracy of 98.0% and an F1-score of 97.73%.

### 4.3. Comparative Analysis and Discussion

#### 4.3.1. Classification Performance for Gait Environments

Figure 6 shows the changes in classification accuracy across the three data partitioning conditions (Cases 1 to 3) as the number of hidden layer units increases from 100 to 1000. In all cases, accuracy generally improved with more hidden layer units. However, Cases 2 and 3 consistently outperformed Case 1, indicating better generalization with more validation data. Additionally, Case 1 exhibited a decrease in accuracy at several points (300, 700, and 900 units), likely due to overfitting from insufficient training data. Conversely, Cases 2 and 3 exhibited relatively stable upward trends in accuracy. These findings suggest that the FFNN effectively learns from multimodal sensor data and accurately classifies different gait environments. Overall, increasing the proportion of validation data improved the stability and predictive accuracy of the model, with a validation split of 15–20% yielding the most consistent results.

The LG environment exhibited the best classification performance, which can be attributed to three possible factors. First, the LG class had the largest data volume, enabling the model to learn more effectively and resulting in higher classification accuracy. Second, because the LG environment represented a common and familiar terrain, participants walked in a more stable and relaxed manner, producing stable physiological signals from the GSR sensor. Finally, the acceleration and angular velocity data in the LG environment exhibited consistent patterns, further aiding model performance.

The proposed FFNN model in this study demonstrated a high overall classification accuracy of approximately 98%. However, there were performance variations across individual classes, with particularly low recall observed for the DR condition. This reduced performance appears to stem from multiple contributing factors. First, the DR dataset was affected by the loss of data from Subject 3 during the collection process, resulting in dataset reduction and class imbalance, both of which likely hindered the model’s learning performance. Second, the IMU sensor attached to the body may not have been sufficiently secured during ramp descent, potentially introducing noise into the data [39,40]. Third, the DR condition shares similar biomechanical movement patterns and sensor characteristics with other environments, such as the AR and stair (AS/DS) conditions. This similarity may have limited the model’s ability to learn subtle distinctions between these environments. Despite the model’s overall high accuracy, these factors contributed to reduced classification performance for specific environments. Future work should consider strategies such as preventing data loss, improving sensor attachment stability, and collecting additional data or extracting environment-specific features to better differentiate between ramp and stair conditions and ultimately improve classification performance.

#### 4.3.2. Performance According to Sensor Combinations

As shown in the results of Section 4.2, the model’s accuracy and F1-score varied depending on the sensor configuration. When using only IMU data, the model achieved the lowest performance, with an accuracy of 92.3% and an F1-score of 92.34%. This suggests that while IMU data alone may be sufficient for basic movement analysis, it lacks the richness of information necessary to distinguish subtle differences between walking environments.

When IMU and smart insole data were combined, the accuracy and F1-score increased slightly by 0.9% and 0.91%, respectively. This improvement can be interpreted as a result of the additional information related to plantar pressure and ground contact, which contributed to a better understanding of gait characteristics. In contrast, the combination of IMU and GSR sensors resulted in an increase of 2.06% in accuracy and 2.00% in F1-score. This indicates that the PPG data obtained through the GSR sensor served as a meaningful signal for distinguishing between different walking environments. PPG, which reflects changes in blood volume, is closely related to heart rate and stress indicators. Physiological responses such as increased heart rate and heightened tension may occur in environments like ramps or stairs, thereby contributing to improved classification performance [41,42]. Therefore, it is likely that differences in physiological loads induced by changes in walking environments were reflected in the data, contributing to the improvement in classification performance. The configuration combining all three sensors (IMU, smart insole, and GSR) achieved the highest performance, with an accuracy of 98.0% and an F1-score of 97.73%. Compared to using the IMU alone, this represents an improvement of 5.7% in accuracy and 5.39% in F1-score. This demonstrates that biomechanical information and physiological response signals functioned complementarily, enabling more precise classification of walking environments. These results support the effectiveness of sensor fusion and highlight the importance of multi-sensor systems in gait analysis [41].

#### 4.3.3. Model Architecture and Real-Time Applicability

The FFNN model proposed in this study has a relatively lightweight architecture, consisting of an input layer, a hidden layer with 1000 neurons, and an output layer. The total number of parameters in the model is 36,001, and the memory usage is approximately 0.137 MB, indicating very low resource consumption. Additionally, the computational cost for inference on a single data sample is approximately 70,000 FLOPS. In previous studies, CNN- and LSTM-based models typically involve hundreds of thousands to millions of parameters, require memory usage in the order of several megabytes, and demand computational costs exceeding several million FLOPS. An LSTM-based gait phase prediction model requires approximately 0.96 million parameters and 3.66 MB of memory [43]. A multimodal CNN model demands an even higher computational load, with 2.59 million parameters and 8.6 G FLOPS [44]. Therefore, the computational efficiency of the proposed model clearly demonstrates an advantage over existing models. This suggests that it can be feasibly implemented in resource-constrained environments requiring real-time processing, such as wearable devices, and is expected to offer significant benefits in terms of practical applicability in real-world settings.

#### 4.3.4. Limitations of the Study

This study had several limitations. First, the small sample size—only five non-disabled participants—may have affected the generalizability of the model. In particular, the data does not reflect the gait patterns of diverse populations such as older adults, individuals with mobility impairments, or actual users of wearable robotic systems. Additionally, there was a noticeable imbalance in the number of samples across walking environments, with level-ground walking data being overrepresented compared to ramp descent data. This imbalance may have introduced a bias in model training and limited the classifier’s ability to equally learn features from all classes. Future studies should include a broader participant base to improve the robustness and applicability of the model.

Second, only the FFNN was employed for classification. Although the FFNN offers advantages in terms of structural simplicity and computational efficiency, we did not conduct direct comparisons with more complex machine learning or deep learning models, such as CNNs or LSTMs, which have been widely used in previous studies using similar sensor modalities (e.g., IMU and insole). This may limit the ability to fully contextualize the performance of our model relative to existing approaches. Therefore, future work should include comparative analyses with alternative models to thoroughly evaluate performance differences and trade-offs.

Third, although a subject-specific cross-validation using the leave-one-subject-out (LOSO) method was conducted to evaluate generalizability, the classification performance decreased compared to that of the random split method. This result highlights the inherent challenge of inter-subject variability and suggests that the model may have limited generalization capability when applied to unseen individuals. This limitation is likely influenced by the small sample size (n = 5), which restricts the diversity of training data in each fold. Future studies should address this issue by including more participants and exploring subject-independent learning strategies.

Fourth, data were collected in controlled indoor settings, which may not reflect the variability of real-world outdoor environments, such as forest trails or gravel surfaces. Consequently, the model may not capture the complex environmental factors encountered in practical wearable system applications.

## 5. Conclusions

This study collected walking data across LG, ramps, and stairs, using an FFNN with a single hidden layer to classify walking environments. Data for ML were obtained from five non-disabled participants using an IMU, smart insole, and GSR, enabling the comprehensive capture of kinematic, biomechanical, and physiological signals. After time synchronization and data filtering, 47,033 processed data samples were used to train and evaluate the model. The proposed approach achieved a classification accuracy of 98%, demonstrating the potential of the FFNN in recognizing walking environments.

Furthermore, the results showed that each sensor modality contributed complementary information to the classification task. While IMU data alone provided basic kinematic features, the addition of smart insole data offered biomechanical insights such as foot pressure and ground contact, and GSR captured physiological responses potentially related to exertion or postural adaptation. The integration of all three sensors yielded the highest performance across all metrics, indicating that multimodal sensor fusion significantly enhances the model’s ability to distinguish between different walking environments. These findings suggest strong potential for real-world application in wearable assistive technologies.

Therefore, future research should aim to improve the generalizability of the dataset by including participants from diverse age groups and with varying gait characteristics. Additionally, various ML algorithms should be compared to identify the most effective classification model. Finally, data should be collected in diverse real-world walking conditions, including outdoor environments, to better assess the practical applicability of the proposed approach.

## Figures and Tables

**Figure 1 sensors-25-04302-f001:**
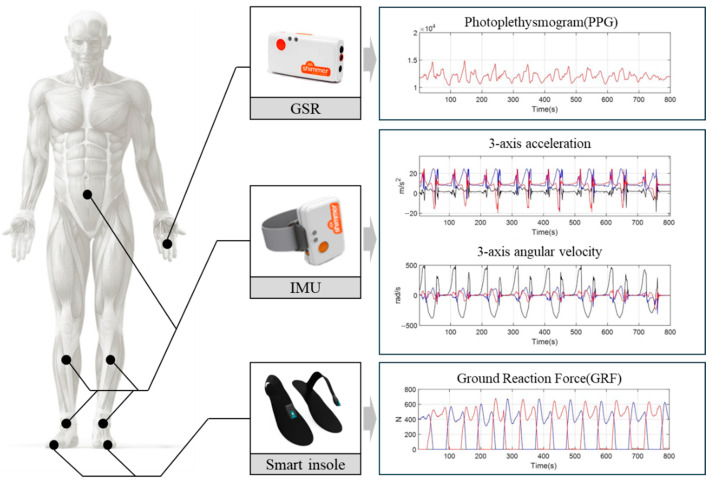
Sensor location and corresponding signal outputs.

**Figure 2 sensors-25-04302-f002:**
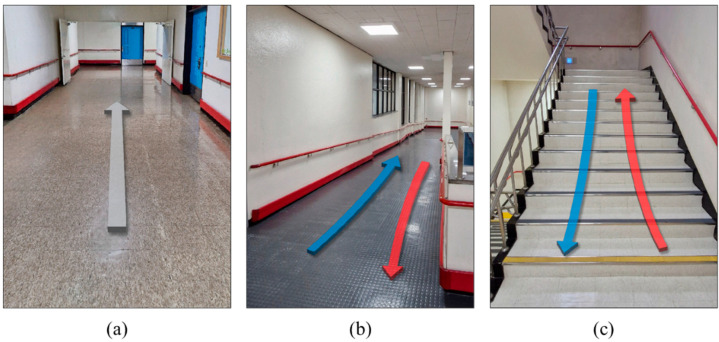
Indoor walking environments used in the experiment: (**a**) LG, (**b**) ascending/descending ramp, and (**c**) ascending/descending stairs.

**Figure 3 sensors-25-04302-f003:**
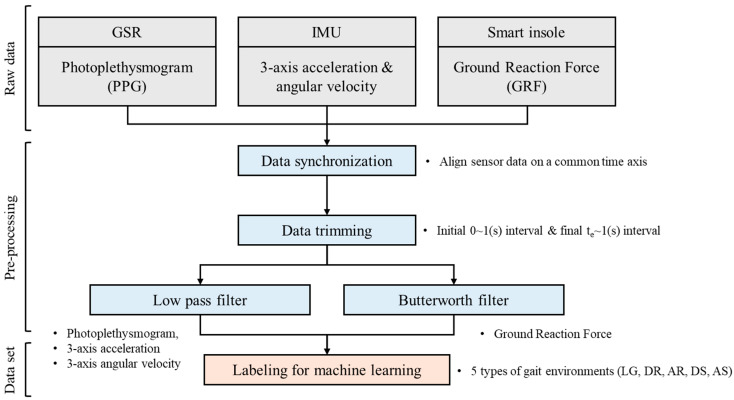
Data processing pipeline for ML based on wearable sensor signals.

**Figure 4 sensors-25-04302-f004:**
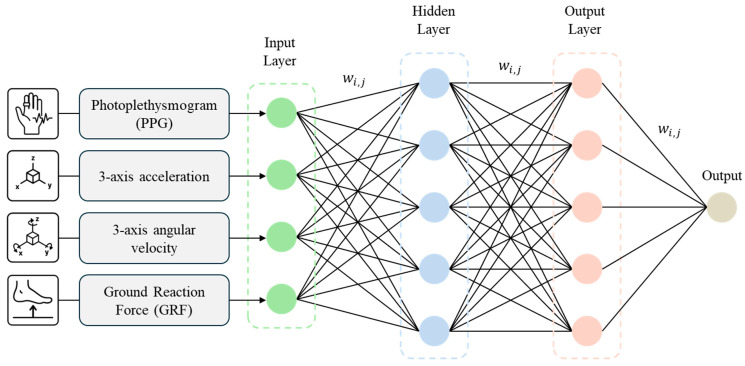
FFNN architecture for multimodal signal-based classification.

**Figure 5 sensors-25-04302-f005:**
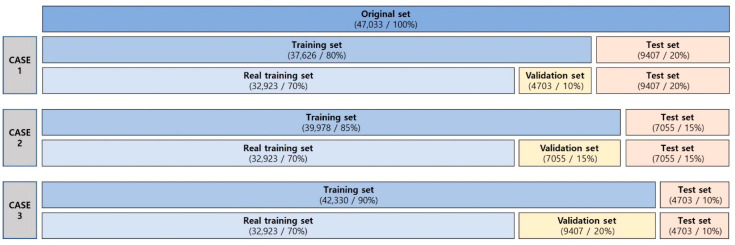
Training, validation, and test set composition for ML classification.

**Figure 6 sensors-25-04302-f006:**
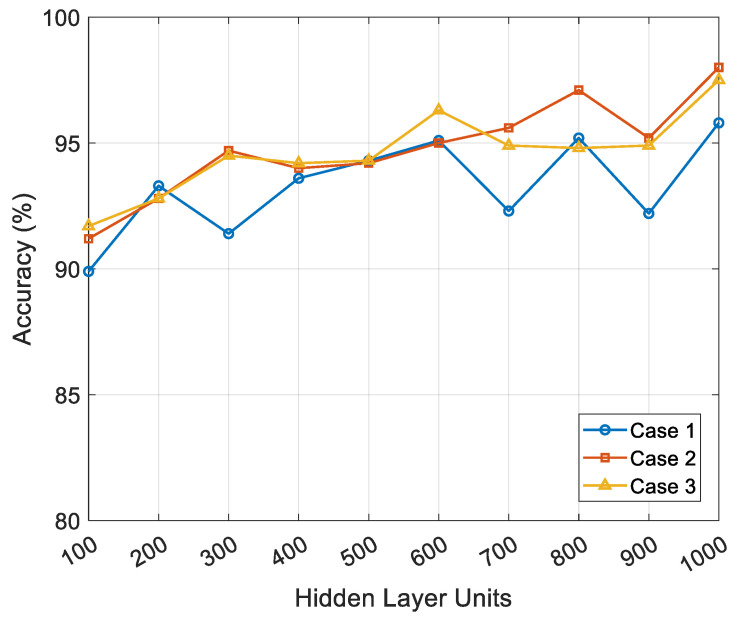
Effect of hidden-layer size and data split ratio on model accuracy.

**Table 1 sensors-25-04302-t001:** Gait data acquired from the sensors.

	LG	DR	AR	DS	AS	Total
S01	2996	2200	3361	767	675	9999
S02	2821	1655	486	1557	1170	7689
S03	3688	-	1364	1456	1655	8163
S04	3127	2245	2244	1853	1857	11,326
S05	3222	1569	1463	1652	1950	9856
Total	15,854	7669	8918	7285	7307	47,033

**Table 2 sensors-25-04302-t002:** Classification accuracy results for each case.

	Hidden Layer Units	Case 1	Case 2	Case 3
		Accuracy (%)		
1st	100	89.9	91.2	91.7
2nd	200	93.3	92.8	92.8
3rd	300	91.4	94.7	94.5
4th	400	93.6	94.0	94.2
5th	500	94.3	94.2	94.3
6th	600	95.1	95.0	96.3
7th	700	92.3	95.6	94.9
8th	800	95.2	97.1	94.8
9th	900	92.2	95.2	94.9
10th	1000	95.8	98.0	97.5

**Table 3 sensors-25-04302-t003:** Confusion matrices for the best-performing cases across the three models.

		Predicted Class						
		LG	DR	AR	DS	AS	Recall	F1-score
Actual Class	LG	15,396	288	299	67	28	95.76%	96.43%
	DR	161	7114	110	47	45	95.15%	93.94%
	AR	183	128	8419	40	52	95.43%	94.92%
	DS	74	63	48	7045	85	96.31%	96.51%
	AS	39	76	42	86	7097	96.69%	96.91%
	Precision	97.12%	92.76%	94.40%	96.71%	97.13%	Macro F1-score	95.83%
(a) Case 1								
		Predicted Class						
		LG	DR	AR	DS	AS	Recall	F1-score
Actual Class	LG	15,638	120	108	35	25	98.19%	98.35%
	DR	89	7432	84	32	27	96.97%	96.68%
	AR	86	65	8666	17	25	97.82%	97.47%
	DS	22	18	23	7158	47	98.49%	98.08%
	AS	39	76	42	86	7097	96.69%	97.48%
	Precision	98.51%	96.38%	97.12%	97.68%	98.28%	Macro F1-score	97.73%
(b) Case 2								
		Predicted Class						
		LG	DR	AR	DS	AS	Recall	F1-score
Actual Class	LG	15,549	184	164	39	29	97.39%	97.73%
	DR	127	7373	74	28	25	96.67%	96.40%
	AR	117	53	8623	30	24	97.47%	97.08%
	DS	37	19	26	7131	47	98.22%	98.05%
	AS	24	40	31	57	7182	97.93%	98.11%
	Precision	98.08%	96.14%	96.69%	97.89%	98.29%	Macro F1-score	97.50%
(c) Case 3								

**Table 4 sensors-25-04302-t004:** Classification performance metrics for different sensor combinations.

Sensor Combination	Accuracy (%)	Precision (%)	Recall (%)	F1-Score (%)
IMU only	92.3	92.24	92.44	92.34
IMU + smart insole	93.2	93.13	93.36	93.25
IMU + GSR	94.4	94.26	94.42	94.34
IMU + smart insole + GSR	98.0	97.60	97.63	97.73

## Data Availability

The data presented in this study are available upon request from the corresponding author owing to privacy concerns and institutional restrictions.

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
