# Peer review of "Gait Environment Recognition Using Biomechanical and Physiological Signals with Feed-Forward Neural Network: A Pilot Study"

_sensors, 2025, doi:10.3390/s25144302_

Round 1
Reviewer 1 Report
Comments and Suggestions for Authors
The manuscript investigates the potential for recognizing terrain conditions encountered by the user through a deep learning-based information fusion approach. The authors have collected both mechanical data and physiological signals, such as PPG, which can provide insight into the user's comfort when navigating different terrains.
The research idea is potentially valid; however, the current approach does not clearly demonstrate the importance of fusing all available information. The authors have not conducted a thorough analysis to understand the individual contribution of each sensor modality to the final decision of the network.
Furthermore, the authors should improve the literature review section, particularly regarding the use of IMU data for gait information reconstruction, as several important and recent studies have not been cited.
Below, I outline my major concerns.
MAJOR CONCERNS
1) In the literature review, Authors should better depict the use of IMU for gait analysis, Indeed IMU can be also used to retrive joint angle displacement which is an important fact. For this reason I suggest Authros to review and report the following reference:
-"Inertial Sensing for Human Motion Analysis: Enabling Sensor-to-Body Calibration Through an Anatomical and Functional Combined Approach." IEEE Transactions on Neural Systems and Rehabilitation Engineering (2025).
2) In the current model development, it is difficult to understand which sensors contribute the most to the final output. Moreover, given the high reported accuracy, it is possible that a reduced sensor setup could achieve comparable results. Based on my experience, using only IMU or GRF data can already yield very high accuracy, potentially eliminating the need for physiological signals such as PPG.
Therefore, I suggest including a pattern recognition experiment in which the authors perform classification using physiological and mechanical data separately, followed by testing all possible sensor combinations. This would help identify the truly optimal sensor configuration. Such an analysis would significantly enhance the scientific contribution of the manuscript and elevate the overall quality of the work.
Author Response
Please refer to the attached document for detailed responses to the reviewer’s comments.

Reviewer 2 Report
Comments and Suggestions for Authors
The paper presents a classification system for gait environments (level ground, ramps, stairs) using multimodal sensor data (IMU, smart insole, GSR) and a feed-forward neural network (FFNN). The topic is relevant for wearable robotics and fall prevention, and the methodology is clearly described. However, several critical issues must be addressed to meet the standards of a Q1 journal like Biosensors:
- The introduction is overly long and includes redundant information, such as extended descriptions of gait biomechanics, EMG functionality, and terrain types, which could be significantly condensed. At the same time, the review of relevant state-of-the-art methods is insufficient, lacking a focused discussion on recent advances in physiological sensing for gait analysis. The paper claims novelty in integrating GSR/PPG signals yet fails to convincingly demonstrate how these signals enhance classification compared to existing IMU- or insole-based approaches. An ablation study would be essential to clarify the actual contribution of PPG. Furthermore, the authors criticize prior work for using complex DL models (e.g., CNNs, LSTMs) that are "unsuitable for real-time systems," but this claim remains unsubstantiated, as no computational benchmarks (e.g., inference time, memory usage) are provided for the proposed FFNN. In addition, since the same sensor modalities (IMU, insole) are employed, the critique of previous studies appears inconsistent without a direct model comparison.
- Claims of the model being "lightweight" and suitable for embedded systems are unsupported. No metrics (e.g., inference time, memory usage) or hardware validation are provided.
- Table 1 reveals critical imbalances (e.g., 15,854 LG samples vs. 7,669 DR samples) and missing data (no DR samples for Subject 3). The authors neither address this nor justify their exclusion.
- The validation method (random split) risks overfitting. Subject-specific cross-validation (e.g., leave-one-subject-out) is needed to prove generalizability.
- The 98% accuracy is misleading without discussion of class-wise performance (e.g., DR recall is lower). Confusion matrices suggest the model struggles with similar terrains (ramps vs. stairs).
- No physiological interpretation of PPG signals is provided. Are they correlated with biomechanical load or stress?
- The limitations section (buried in conclusions) should be expanded and moved to the discussion. Key issues (small sample size, indoor-only data) need deeper critique.
- Many references are from 2015–2021, which is problematic in a rapidly evolving field like wearable sensing and deep learning. Recent works (2023–2025), especially those involving multimodal data and lightweight models, should be incorporated to better position the study.
Author Response

(The authors gave the same response as above.)

Round 2
Reviewer 1 Report
Comments and Suggestions for Authors
Authors addressed my concerns.
Reviewer 2 Report
Comments and Suggestions for Authors
The authors have addressed the concerns expressed in the initial review, and illustrated that in the paper.
I would have expected thus also some other improvements in the approaches and methods used to analyise the data, not only the validation of the raised concerns.
However all the problems were addressed so I will leave the final decision to the editor.